# Synthesis and Characterization of Novel Thiosalicylate-based Solid-Supported Ionic Liquid for Removal of Pb(II) Ions from Aqueous Solution

**DOI:** 10.3390/molecules28020830

**Published:** 2023-01-13

**Authors:** Nur Anis Liyana Kamaruddin, Mohd Faisal Taha, Cecilia Devi Wilfred

**Affiliations:** 1Centre of Research in Ionic Liquids, Universiti Teknologi PETRONAS, Seri Iskandar, Perak 32610, Malaysia; 2Fundamental and Applied Sciences Department, Universiti Teknologi PETRONAS, Seri Iskandar, Perak 32610, Malaysia

**Keywords:** ionic liquids, solid-supported ionic liquids, thiosalicylate, Pb(II) removal

## Abstract

The main objectives of this study are to synthesize a new solid-supported ionic liquid (SSIL) that has a covalent bond between the solid support, i.e., activated silica gel, with thiosalicylate-based ionic liquid and to evaluate the performance of this new SSIL as an extractant, labelled as Si-TS-SSIL, and to remove Pb(II) ions from an aqueous solution. In this study, 1-methyl-3-(3-trimethoxysilylpropyl) imidazolium thiosalicylate ([MTMSPI][TS]) ionic liquid was synthesized and the formation of [MTMSPI][TS] was confirmed through structural analysis using NMR, FTIR, IC, TGA, and Karl Fischer Titration. The [MTMSPI][TS] ionic liquid was then chemically immobilized on activated silica gel to produce a new thiosalicylate-based solid-supported ionic liquid (Si-TS-SSIL). The formation of these covalent bonds on Si-TS-SSIL was confirmed by solid-state NMR analysis. Meanwhile, BET analysis was performed to study the surface area of the activated silica gel and the prepared Si-TS-SSIL (before and after washing with solvent) with the purpose to show that all physically immobilized [MTMSPI][TS] has been washed off from Si-TS-SSIL, leaving only chemically immobilized [MTMSPI][TS] on Si-TS-SSIL before proceeding with removal study. The removal study of Pb(II) ions from an aqueous solution was carried out using Si-TS-SSIL as an extractant, whereby the amount of Pb(II) ions removed was determined by AAS. In this removal study, the experiments were carried out at a fixed agitation speed (400 rpm) and fixed amount of Si-TS-SSIL (0.25 g), with different contact times ranging from 2 to 250 min at room temperature. The maximum removal capacity was found to be 8.37 mg/g. The kinetics study was well fitted with the pseudo-second order model. Meanwhile, for the isotherm study, the removal process of Pb(II) ions was well described by the Freundlich isotherm model, as this model exhibited a higher correlation coefficient (R^2^), i.e., 0.99, as compared to the Langmuir isotherm model.

## 1. Introduction

Fast evolution in industrial enterprise and poor wastewater treatment has led to the discharge of a tremendous amount of waste, which contains hazardous chemicals and expels pollutants into the environment. Thus, it is critically important to remove or reduce the amounts of heavy metals in wastewater to an allowable safe limit before discharging them into the natural environment [1]. One of the common hazardous metal ions present in wastewater, such as in the petroleum industry, is the Pb(II) ion. According to Akpoveta et al., in the petroleum industry, Pb(II) ions are found because of the chemicals used during the refining process, metals absorption coming from pipelines, and vessels and tanks, together with the natural metals, existing in sandstone during the crude extraction process [2]. Pb(II) ions can be absorbed and accumulated in the human body and cause serious health effects such as cancer, causing damage to the kidneys, liver, heart, brain, bones, and neurological system in humans through inhaling and swallowing polluted food and water [3]. They represent serious threats to the human population and the fauna and flora of the receiving water bodies, as the low amounts of this heavy metal are highly toxic. Hence, it is very crucial to remove Pb(II) ions from wastewater and the environment.

Several commercial methods are being used in wastewater treatment to remove metal ions, namely, reverse osmosis, ion exchange, chemical precipitation, irradiation, biosorption, and extraction [4]. Reverse osmosis can remove many types of molecules and ions from solutions, including bacteria, and is used in both industrial processes. Reverse osmosis involves a diffusive mechanism, so the separation efficiency is dependent on a solute concentration, pressure, and water flux rate [5]. Ion exchange can attract soluble ions from the liquid phase to the solid phase, which is the most widely used method in the water treatment industry. As a cost-effective method, the ion exchange process normally involves low-cost materials and convenient operations. However, it is very sensitive to the pH of an aqueous solution and can be used only at low concentrated metal solutions [6]. Chemical precipitation is widely used because of its simple operation [7]. However, it requires a large amount of chemicals to reduce metals to an acceptable level for discharge. Other drawbacks are huge sludge production, slow metal precipitation, poor settling, the aggregation of metal precipitates, and the long-term environmental impacts of sludge disposal. Another method that has gained attention during recent years for heavy metal removal is the usage of plant biomass as the extractant through the biosorption process, owing to its good performance and being a low-cost material [8]. Although biological methods are inexpensive, environmentally friendly techniques, they need large areas and proper maintenance and operation [5].

Liquid–liquid extraction or solvent extraction is one of the frequently used methods in several industries such as electronic and battery industries to remove metal ions from their wastewater. Liquid–liquid extraction may be an excellent option for wastewater treatment as it offers several advantages such as high extractability and high selectivity for liquid separation [9]. However, there are a few drawbacks to liquid–liquid extraction, such as emulsion formation, the usage of huge volatile organic solvents that are carcinogenic and non-biodegradable and, hence, the formation of large amounts of pollutants, making it costly, time-consuming, and ecologically unfriendly [10]. The problems associated with the usage of volatile organic solvents in liquid–liquid extraction have motivated many researchers to find alternative metal extractants that would be efficient, cost effective, and environmentally friendly. One of the alternatives that is being extensively studied by researchers to replace organic solvents as metal extractants in liquid–liquid extraction is ionic liquids (ILs) [11].

Ionic liquid mainly consists of an organic cation with an organic or inorganic anion, which is a liquid organic salt that is bulky in chemical structure [12]. ILs are salts and generally exist in liquid form at temperatures below 100 °C. The physical and chemical properties of ILs can be altered as desired by modifying the cation and anion combinations [13]. Because of their excellent tunability, ILs have great potential to form different combinations in the designing of task-specific fluids. This feature would give access to more applications of ILs, such as replacing common volatile organic solvents with ILs in the liquid–liquid extraction method, which allows for the development of more effective separation processes. The substitution of volatile organic solvents with ILs also prevents organic solvent losses by evaporation; hence, resulting in material loss and reducing environmental impact [14]. Furthermore, ILs have unique properties which make them different from ordinary salts, such as being non-volatile, non-flammable solvents that have a negligible vapor pressure [15].

Aside from the fact that ILs are considered as greener solvents as compared to volatile organic solvents, they also have limitations such as being expensive, having high viscosity, high cost separation, and being faced with the difficulty to maintain extracted analytes in their phase (mass transfer problem) [16]. To cope with these problems, a method based on immobilization or impregnation of ionic liquid on a solid support as a solid phase extractor, known as a solid-supported ionic liquid (SSIL), is being studied extensively by many researchers [17,18,19]. This immobilization method producing SSIL creates a thin layer of ionic liquid on a solid support which in turn decreases the amount of ionic liquid required to extract metal ions compared to using ionic liquid in liquid–liquid extraction [20].

The removal studies have shown that SSIL performed better in removing metal ions from aqueous solution as compared to well-known solid acids such as silica, alumina, and zeolites. This is caused by the high surface area for contact between the ionic liquid (contain functional group(s) having high affinity towards metal ions) immobilized on a solid support in SSIL with the targeted metal ions. In addition to offering different functional groups in ionic liquid immobilized on its solid support, the acidity of SSIL can be tunable depending on the choice of solid support [21]. In fact, SSIL is easy to apply for a large scale of operation. Moreover, SSIL can be successfully employed in both batch and flow processes [20].

In this piece of work, an attempt was made to chemically immobilize 1-methyl-3-(3-trimethoxysilylpropyl) imidazolium thiosalicylate ([MTMSPI][TS]) ionic liquid on activated silica gel to produce a new solid-supported ionic liquid known as thiosalicylate-based solid-supported ionic liquid (Si-TS-SSIL). The [MTMSPI][TS] ionic liquid containing the thiosalicylate functional group was chosen to be chemically immobilized on activated silica gel in SSIL (as an extraction agent for Pb(II) ions from aqueous solution) because of the high affinity of thiosalicylate towards metal ions [21]. Meanwhile, Pb(II) ions have been selected as this metal ion is commonly found in industrial wastewater [11]. The kinetics studies for the removal of Pb(II) ions was carried out at different contact times (2–250 min). As for the adsorption isotherms, this study was conducted to elaborate the insight of the adsorption process with different initial Pb(II) concentrations ranging from 10–200 ppm. For the first time, the newly synthesized Si-TS-SSIL extractant in the removal of Pb(II) ions in an aqueous solution and wastewater were performed. It is an advanced creation that could potentially provide both low-cost and efficiency compared to the sole usage of ILs as metal extractants. Therefore, this current study has great potential to contribute to wastewater and environmental cleanliness. In addition, this adsorption method is easy to apply and has a short test period.

## 2. Results and Discussions

### 2.1. Characterization

The ^1^H-NMR spectra of the synthesized [MTMSPI][TS] ionic liquid is depicted in Figure 1. As shown by Figure 1, chemical shifts between 6.8 to 7.6 ppm could clearly be assigned to aromatic hydrogen atoms of the thiosalicylate anion [22]. The ^1^H-NMR of thiosalicylate anion in [MTMSPI][TS] is summarized as: δ 6.89 (2H, CH), δ 7.13 (1H, CH), δ 7.6 (1H, CH). Meanwhile, the ^1^H-NMR of 1-methyl-3-(3-trimethoxysilylpropyl) imidazolium cation in [MTMSPI][TS] is summarized as: δ 0.55 (2H, CH_2_), δ 1.91 (2H, CH_2_), δ 3.66 (9H, CH_3_), δ 3.94 (3H, CH_3_), δ 4.27 (2H, CH_2_), δ 7.84 (2H, CH), δ 9.78 (1H, CH). The chemical shift for the NMR solvent, i.e., dimethyl sulfoxide (DMSO), is 2.5 ppm.

The FTIR spectra of the synthesized [MTMSPI][TS] ionic liquid was recorded by Thermo Scientific spectrometer using the attenuated total reflectance (ATR) method and is shown in Figure 2. In activated silica gel, the silanol group was observed by the presence of broad -OH stretch at 3425 cm^−1^. The siloxane group (Si-O-Si) asymmetric stretching was appeared at 1081 cm^−1^ and the corresponding symmetric stretching was observed at 793 cm^−1^ [23].

In the [MTMSPI][TS] ionic liquid, the absorption band at 2942 cm^−1^ was indicated to the stretching mode of -CH_2_ groups, which were related to trimethoxysilylpropyl. Si-O-Si tensile vibration was observed at 1034 cm^−1^ [24]. Another absorption band appeared at 1571 cm^−1^ because of C-N stretching [25]. Additionally, a weak peak detected around 2232 cm^−1^ belongs to the S-H group of the aromatic compound in thiosalicylate ion of [MTMSPI][TS] [26].

For Si-TS-SSIL, which contains the combination of activated silica gel and [MTMSPI][TS] ionic liquid, stretching vibrations of the Si-O-Si groups were observed at 1030 cm^−1^ [23]. The peaks observed in the range of 1550–1650 cm^−1^ were related to the C-N stretching of the imidazolium ring. Meanwhile, the O-H vibration observed in the silica gel particle will be decreased when it is modified with ionic liquid. It can be seen in the figure below where the lack of peak in the range of 3300–3500 cm^−1^ is significant. This indicates that the imidazolium cation is in interaction with the activated silica gel [27]. The presence of the expectable functional groups in the prepared materials at the respective position on the FTIR spectrum indicates the successful anchorage of the organic ligands and alkyl silanes onto the silica framework.

Meanwhile, the moisture content of the [MTMSPI][TS] ionic liquid was below 10.00 ppm. Since sodium chloride (NaCl) dissolved in the solvent is the by-product for synthesizing [MTMSPI][TS], the chloride content was determined as part of the purity study. As shown in Table 1, the chloride content in [MTMSPI][TS] was found to be 9.08 ppm. With regards to chloride content, the purity of synthesized [MTMSPI][TS] was calculated to be 99.91%.

The thermogravimetric analysis of the synthesized [MTMSPI][TS] was studied in the range of 100 °C to 700 °C to observe its thermal stability. As shown in Figure 3, three characteristic decomposition stages were observed. The first TGA curve shows a mass loss of about 9.09% up to 196 °C because of the removal of adsorbed water molecules [28]. The second weight loss occurred from 224 °C to 401 °C, which was assigned to the degradation of thiol group [29]. Further weight loss (8.19%) that was noticed from 417 °C to 532 °C could be associated with the degradation of the remaining organic molecules [30]. From the TGA analysis, the newly synthesized [MTMSPI][TS] ionic liquid exhibited high thermal stability.

The newly synthesized solid-supported ionic liquid with a thiol functional group, Si-TS-SSIL, was analyzed using solid-state NMR with the purpose to confirm the formation of covalent bonds (Si-O bonds) between [MTMSPI][TS] and activated silica gel in Si-TS-SSIL, i.e., the covalent between the cation of [MTMPSI][TS] and the silanol group (Si-OH) on activated silica gel. The Si-OH group dominates the surface properties of activated silica gel, and this group corresponds to the cation of [MTMSPI][TS], 1-methyl-3-(3-trimethoxysilylpropyl) imidazolium, to perform chemical immobilization. In the solid-state NMR analysis, the solid-state of silica cross-polarization magic angle spinning (^29^Si CP-MAS) NMR spectroscopy was utilized to show that [MTMSPI][TS] was covalently bonded to activated silica gel in Si-TS-SSIL. As shown in Figure 4a, the ^29^Si MAS NMR spectrum of pure activated silica gel revealed the presence of three signals with resolved peaks at −91, −100, and −109 ppm. These peaks are assigned to silicon atoms in the silanediol groups (Q^2^), silanol groups (Q^3^) and silicon-oxygen tetrahedra (Q^4^) of the SiO_2_, respectively [31]. After chemical modification of activated silica gel surface with the [MTMSPI][TS] ionic liquid producing Si-TS-SSIL, an increase in the intensity growth was observed for signal Q^4^ followed by a significant reduction in the signal Q^2^ and Q^3^ [32]. Additionally, the appearance of new peaks was found, which appeared at −57 and −66 ppm of the T^2^ and T^3^ units as shown in Figure 4b. Both peaks are attributed to the silanisation of the silica particles surface, which produced the covalent bonds of Si-O, thus, confirmed that [MTMSPI][TS] covalently bonded to the activate silica gel surface in Si-TS-SSIL [33].

The surface area, pore volume, and pore diameter of activated silica gel and Si-TS-SSIL (before and after washing with dichloromethane (solvent)) were determined using BET. Based on Table 2, the activated silica gel and Si-TS-SSIL could be classified into the category of mesoporous as their pore size is in the range of 2–10 nm [34]. The surface area, pore volume and pore diameter of Si-TS-SSIL, for both samples of before and after washing with dichloromethane, were reduced as compared to activated silica gel. These results could be explained by the fact that the immobilization, either physically or chemically, of [MTMSPI][TS] on activated silica gel would reduce the surface area, pore volume, and pore diameter of Si-TS-SSIL. After washing with dichloromethane, as expected, the surface area, pore volume, and pore diameter of Si-TS-SSIL were increased compared to the one before washing with dichloromethane. These achieved results might be caused by the removal of [MTMSPI][TS], which was physically immobilized on the activated silica gel in Si-TS-SSIL by dichloromethane, which in turn left only chemically immobilized [MTMSPI][TS]. In this study, it is important to remove physically immobilized [MTMSPI][TS] on activated silica gel in Si-TS-SSIL to prevent the leaching of [MTMSPI][TS] during the removal study.

### 2.2. Adsorption Studies

#### 2.2.1. Effect of [MTMSPI][TS]: Activated Silica gel Mass Ratio in Si-TS-SSIL on the Removal Efficiency

In this piece of work, an attempt was first made to analyze the performance of three control samples of extractant ([MTMSPI][TS], activated silica gel, and Si-TS-SSIL) to remove Pb(II) ions from the aqueous solution with the following constant parameters: initial Pb(II) ions concentration, with mixing time and extractant dosage being 200 mg L^−1^, 30 min, and 0.25 g, respectively. The results obtained are shown in Table 3. As expected, Si-TS-SSIL exhibited better removal efficiency compared to activated silica gel, but lower performance compared to [MTMSPI][TS]. With this confirmation on the results of removal efficiency by Si-TS-SSIL, the removal study proceeded using four samples of Si-TS-SSIL as an extractant.

Four samples of Si-TS-SSIL that have a different mass ratio of [MTMSPI][TS] and activated silica gel, i.e., [MTMSPI][TS](in g):activated silica gel (in g), were analyzed on the perspective of removal efficiency. The purpose of this analysis is to study the dosage effect of [MTMSPI][TS] in Si-TS-SSIL in removing Pb(II) ions from the aqueous solution. Figure 5 depicts the relationship between the Si-TS-SSIL’s removal capacity and the [MTMSPI][TS]:activated silica gel mass ratio in Si-TS-SSIL to remove Pb(II) ions from the aqueous solution. From this figure, it can be seen that the increasing amount of [MTMSPI][TS] resulted in an increase of removal efficiency until the system reached equilibrium, when the [MTMSPI][TS]:activated silica gel mass ratio was 0.2:1, whereby there was no significant difference in removal efficiency beyond the mass ratio of 0.2:1. Thus, further experiments of removal study were conducted based on 0.2:1 ([MTMSPI][TS]:activated silica gel) mass ratio as it was the optimum mass ratio for the removal process of Pb(II) ions from an aqueous solution using Si-TS-SSIL as an extractant.

#### 2.2.2. Effect of Contact Time

In the batch removal process, contact time is one of the most important factors as it defines the time required for the extractant to reach a dynamic equilibrium stage. Temperature (25 °C), Si-TS-SSIL dosage (0.25 g), initial concentration Pb(II) ions (200 mg L^−1^), and agitation speed (400 rpm) were all held constant at this step, except for the contact time. Figure 6 depicts the effect of contact time on Pb(II) ions removal efficiency using an [MTMSPI][TS]:activated silica gel mass ratio of 0.2:1 in Si-TS-SSIL. As can be seen in Figure 6, the removal rate grew rapidly at first and was followed by a subsequent slow uptake. The optimum removal efficiency of Pb(II) ions was established in approximately 120 min. These results of the removal process could be explained, wherein when the number of accessible sites (functional group, i.e., thiol group, and number of pores) is substantially greater than the number of metal species to be removed, the removal process appears to proceed rapidly. As the contact time increased, the amount of Pb(II) ions removed also increased up to a certain contact time, whereby the removal phase was reached at a steady state. Thus, in this removal study, the optimum contact time was decided at 120 min for the removal of Pb(II) metal ions from the aqueous solution with 87% removal efficiency.

#### 2.2.3. Effect of pH

The pH of the aqueous solution from which Pb(II) ions were removed significantly affected the extraction process because of the active sites and charge distribution on the Si-TS-SSIL surface, solubility, ionization, and speciation of Pb(II) ions in the solution. In this study, the effect of pH on Pb(II) metal removal in an aqueous solution was investigated between pH 3 to 9 using an Si-TS-SSIL extractant. The removal efficiency of the Si-TS-SSIL extractant increased with an increase in the pH with highest removal efficiency observed at pH 6, as shown in Figure 7. Above this value, the amount of Pb(II) metal ions removed decreased due to possible precipitation of Pb(II) hydroxide. In an acidic condition, the removal efficiency is low, which could be caused by the presence of protonated functional groups containing lone pairs [35].

### 2.3. Adsorption Kinetics

One of the essential characteristics that determines the extraction efficiency is the kinetic of the removal process, which describes the rate of removal of Pb(II) ions. The results of the kinetic analysis were used to establish the optimum mechanism for Pb(II) ions removal. Table 4 provides the summary of all the relative values and the statistical analysis for all models, with the most significant values highlighted. Pseudo-first order and pseudo-second order were applied in this kinetic study to understand the dynamics extraction of Pb(II) ions onto an Si-TS-SSIL extractant. The linearized forms of pseudo-first order and pseudo-second order kinetic models are presented in the Equations (1) and (2), respectively [36].

(1)$$\log(q_{e}-q_{t})=\log q_{e}-(\frac{k_{1}}{2.303})t$$(2)$$\frac{t}{q_{t\;}}=\frac{1}{k_{2\;}q_{e\;}{}^{2}}+\left(\frac{1}{q_{e\;}}\right)t$$ where *q_t_* and *q_e_* are the amount of the Pb(II) ions removed (mg g^−1^) at any time and the amount of Pb(II) ions removed (mg g^−1^) at equilibrium, respectively. The pseudo-first order rate constant (min^−1^) is represented as *k_1_* whereas the rate constant for pseudo-second order (g mg^−1^ min^−1^) is *k_2_*. The rate constant and removal capacities for both models were calculated from the slope and intercept of the graphs (Figure 8 and Figure 9).

**Table 4 molecules-28-00830-t004:** Kinetics model for the removal of Pb(II) ions by the Si-TS-SSIL extractant.

Extractant	Pseudo-First Order			Pseudo-Second Order		
Si-TS-SSIL	*q_e exp_* (mg/g)	*q_e cal_* (mg/g)	*R^2^*	*q_e exp_* (mg/g)	*q_e cal_* (mg/g)	*R^2^*
	8.3970	5.5182	0.8705	8.3970	8.9718	0.9942

To identify the best fit of kinetic models for the removal of Pb(II) ions, the values of the correlation coefficient (*R^2^*) of the linear plots and the calculated removal capacities (*q_e calc_*) against the experimental removal capacities (*q_e exp_*) were compared between the pseudo-first and second order kinetic models. From this comparison, the value of *R^2^* for pseudo-second order reaction (0.9942) was higher compared to the pseudo-first order (0.8705). Moreover, the calculated removal capacities (*q_e calc_*) for the pseudo-second order, which is 8.9718 mg g^−1^, was found to be very close to the experimental values (8.397 mg g^−1^). Meanwhile, the *q_e calc_* for pseudo-first order was 5.5182 mg g^−1^, which was significantly different from the experimental removal capacity. Thus, this study suggested that that the pseudo-second order reaction better represents the uptake of Pb(II) ions onto an Si-TS-SSIL extractant from an aqueous solution. These results also show that the removal process was controlled by the chemisorption process [22].

### 2.4. Adsorption Isotherm

The experimental data of isotherms study has been described using a variety of adsorption models. The most used models are the Langmuir and Freundlich models. Both models were employed in this study to analyse the best equilibrium interaction. The Langmuir isotherm theory assumes that a monolayer adsorbate is formed on a homogeneous adsorbent surface. The Langmuir isotherm predicts monolayer adsorption on a homogenous surface with a limited amount of adsorption sites, with no intermolecular interactions occurring between the adsorbed molecules [37]. The Langmuir isotherm equation relates the amount of adsorbate adsorbed on the adsorbent to the equilibrium concentration, which is shown as follows in Equation (3):
(3)$$\frac{C_{e}}{q_{e}}=\frac{C_{e}}{q_{m}}+\;\left(\frac{1}{K_{L}\;\cdot \;q_{m}}\right)$$ where *q_e_* is the amount of Pb(II) ions removed by the Si-TS-SSIL extractant, *KL* and *q_m_* are the Langmuir constants, representing the energy constant (L mg^−1^) and the maximum removal capacity (mg g^−1^), respectively. From the graph in Figure 10, it was found that the correlation coefficient, *R^2^* of the Langmuir isotherm, was 0.9102. The Langmuir equation can also be used to determine a dimensionless equilibrium parameter called the separation factor, *R_L_* which is mathematically defined in Equation (4).

(4)$$R_{L}=\frac{1}{\left(1+K_{L}\;\cdot \;C_{o}\right)}$$ where *K_L_* and *C_o_* are Langmuir constant (L mg^−1^) and initial concentration of Pb(II) ions (mg L^−1^), respectively. The linear removal process is represented by *R_L_* = 1, whereas the irreversible removal process is represented by *R_L_* = 0. Favourable removal is represented by 0 < *R_L_*< 1, whereas unfavorable removal is represented by *R_L_* > 1 [38]. In this case, the *R_L_* value was calculated and was found to be 0.3808. This result indicated that the removal of Pb(II) ions onto the Si-TS-SSIL extractant is a favourable removal process.

The Freundlich equation is expressed by Equation (5).
(5)$${\mathrm{lnq}}_{e}=\frac{1}{n\;}\left(\frac{ln}{Ce}\right)+\ln K_{F}$$ where *q_e_* is the Pb(II) ions uptake capacity (mg g^−1^), *C_e_* is the residual concentration of Pb(II) ions at equilibrium (mg L^−1^), and *n* and *K_F_* are the Freundlich constants, representing the removal intensity and removal capacity (mg g^−1^), respectively. A greater *n* value (*n* > 1) suggests better removal performance, whereas *n* < 1 indicates poor removal performance [25]. In this study, as shown in Table 5, the value of *n* indicates a favourable removal process. The most prevalent condition is *n* > 1, which can be caused by a distribution of surface sites or any other circumstance that causes a decrease in adsorbent–adsorbate interaction as surface density rises. From Figure 10 and Figure 11, the isotherm data fitted well with the Freundlich model with correlation coefficient (*R^2^*) of 0.9961 compared to the Langmuir isotherm model.

**Figure 10 molecules-28-00830-f010:**
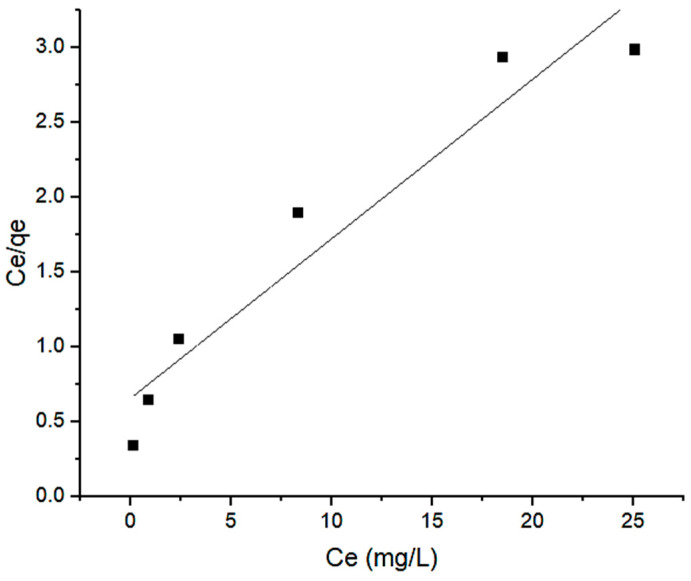
Langmuir isotherm for the removal of Pb(II) ions from aqueous solution by Si-TS-SSIL.

### 2.5. Adsorption Mechanism

For the adsorption mechanism using Si-TS-SSIL, the electron donors coming from the hydroxyl group, and the oxygen and sulphur of the thiosalicylate functional group in Si-TS-SSIL could be responsible for the chemisorption of Pb(II) ions. In addition to this, the organic and inorganic constituents’ presence in the Si-TS-SSIL extractant offered more active sites to bind with Pb(II) ions in the aqueous solution [39]. Additionally, Pb(II) ions would have been trapped in the pores of Si-TS-SSIL, which could contribute to the removal efficiency. Therefore, greater removal efficiency of Pb(II) ions by Si-TS-SSIL could be caused by the thiol functional group existing on Si-TS-SSIL extractant surface for binding.

### 2.6. Adsorption from Wastewater

The adsorption performance of the Si-TS-SSIL extractant was evaluated by the experiments carried out using crude oil effluent containing 10 mg L^−1^ of Pb(II) ions. In this test, 0.25 g of Si-TS-SSIL extractant was dispersed into 12 mL of the wastewater samples, and the mixture was shaken for 120 min. It was found from the obtained results that Pb(II) was removed by 99%. Table 6 below shows the comparison of the removal efficiency of Pb(II) ions using Si-TS-SSIL from an aqueous solution and wastewater. From these results, Si-TS-SSIL can potentially be used as a metal extractant for the treatment of wastewater containing Pb(II) ions.

## 3. Materials and Methods

### 3.1. Chemicals

Silica gel, sodium hydroxide pellets (98% purity), hydrochloric acid fuming 37%, diethyl ether, methanol, toluene, dichloromethane, and acetonitrile were acquired from Merck, NJ, USA. Meanwhile, 1-methylimidazole, 3-chloropropyltrimethoxysilane (CPTMS), thiosalicylic acid (97% purity), and Pb(II) nitrate salt (99.99% purity) were purchased from Sigma-Aldrich, St. Louis, MO, USA.

### 3.2. Instrumentation

A nuclear magnetic resonance (NMR) instrument (Model 1200 Series Avance III, Bruker, Billerica, MA, USA) was used to obtain the ^1^H-NMR spectra of newly synthesized [MTMSPI][TS] ionic liquid, and to perform solid-state NMR analysis to confirm the formation of the covalent bond formed between [MTMSPI][TS] and activated silica gel in a new thiosalicylate-based solid-supported ionic liquid (Si-TS-SSIL). The functional groups of the [MTMSPI][TS] ionic liquid were determined using Fourier-transform infrared spectrophotometer (FTIR) instrument (Model Frontier, Thermo Fisher Scientific, Waltham, MA, USA). A Karl Fischer titrator (Model KF V30, Mettler Toledo, Columbus, OH, USA) was used to analyze the water content in the [MTMSPI][TS] ionic liquid. A thermal analyzer (Model STA 6000, Perkin Elmer, Waltham, MA, USA) was used to record the thermal stability whereas the halide content in [MTMSPI][TS] ionic liquid was analyzed using an ion chromatography (IC) instrument (Model Compact 930, Metrohm, Herisau, Switzerland). The surface area of Si-TS-SSIL was measured using a Brunauer-Emmett-Teller (BET) instrument (Tristar 3020, Micrometritics, Norcross, GA, USA). The concentration of Pb(II) ions in an aqueous solution was determined by an atomic absorption spectrometer (AAS) from Agilent (Model 240 FS, Agilent Technologies, Santa Clara, CA, USA).

### 3.3. Procedures

#### 3.3.1. Synthesis of [MTMSPI][TS] Ionic Liquid

1-methyl-3-(3-trimethoxysilylpropyl) imidazolium thiosalicylate ([MTMSPI][TS]) ionic liquid was synthesized in two steps. The first step is the synthesis of the reagent required to produce [MTMSPI][TS], i.e., the 1-methyl-3-(3-trimethoxysilylpropyl) imidazolium chloride ([MTMSPI][Cl]) ionic liquid. A-4.13 g (25 mmol) of 1-methylimidazole was added to a stirred solution of 3-chloropropyltrimethoxysilane (25 mmol) to form [MTMSPI][Cl] ionic liquid. This mixture was refluxed under 80 °C and stirred at 400 rpm for 72 h in the silicone oil bath. The product, i.e., [MTMSPI][Cl], was decanted with diethyl ether three times to remove impurities and unreacted starting materials [40] and the remaining solvent was extracted by using a rotary evaporator.

Thereafter, in the second step, sodium thiosalicylate (25 mmol) was added to the obtained [MTMSPI][Cl] and stirred for 48 h at room temperature to complete the anion exchange process to obtain the new [MTMSPI][TS] ionic liquid with the by-product of NaCl. The by-product salt, NaCl, was filtered after the addition of acetonitrile and the solvent was then removed by rotary evaporator. The [MTMSPI][TS] ionic liquid that was obtained was found to be a clear and viscous yellow liquid and this newly synthesized ionic liquid was dried under high vacuum. The synthesis route for the formation of [MTMSPI][TS] ionic liquid is shown in Figure 12. The newly synthesized [MTMSPI][TS] ionic liquid was analyzed for structural analysis using NMR and FTIR. The purity of this newly synthesized ionic liquid was also analyzed in terms of halide content and moisture content using IC and the Karl Fischer titrator, respectively. Meanwhile, the thermal analysis including the decomposition temperature of [MTMSPI][TS] was studied using TGA.

#### 3.3.2. Preparation of Solid-Supported Ionic Liquid Containing the Thiosalicylate Functional Group (Si-TS-SSIL)

Si-TS-SSIL was prepared through covalently immobilized [MTMSPI][TS] onto activated silica gel. This preparation involved two steps. The first step was to activate the silica gel through acid activation to obtain activated silica gel surfaces, which is to activate surface silanol groups. These silanol groups would make covalent bonds with the cation of [MTMSPI][TS] to form a new solid-supported ionic liquid solid (Si-TS-SSIL). In this step, activated silica gel (10 g) was immersed in 100 mL of 6 M HCl. The mixture was then refluxed and stirred for 8 h. The suspension was filtered and washed several times with distilled water and ethanol to remove any HCl residue. It was then dried in a vacuum oven for about five hours at the temperature of 70 °C to remove any excess moisture to obtain the activated silica gel.

In the second step, 1 g of the synthesized [MTMSPI][TS] was dissolved in methanol (20 mL) and toluene (20 mL), followed with the addition of 6 g of activated silica gel. This mixture was left stirred and refluxed at 100 °C for 48 h and Si-TS-SSIL was vacuum-filtered and washed with dichloromethane to ensure the complete elimination of solvent. Si-TS-SSIL was then dried in a vacuum oven at 120 °C overnight. The synthesis route to prepare Si-TS-SSIL is shown in Figure 13.

Based on the second step, an attempt was also made to prepare Si-TS-SSIL with a different mass ratio of [MTMSPI][TS] and activated silica gel, i.e., [MTMSPI][TS](in g):activated silica gel (in g). The purpose of this attempt is to study the effect of different amounts of [MTMSPI][TS] covalently bonded onto activated silica gel in Si-TS-SSIL on the removal efficiency of Pb(II) ions from an aqueous solution. The mass ratios of ([MTMSPI][TS](in g): activated silica gel (in g)) studied were 0.1:1, 0.12:1, 0.2:1, and 0.5:1.

#### 3.3.3. Removal Study

An attempt was first made to compare the removal efficiency between three extractant, i.e., [MTMSPI][TS] ionic liquid, activated silica gel, and Si-TS-SSIL, to remove Pb(II) ions from an aqueous solution. In this removal study, 200 mg L^−1^ of aqueous Pb(II) ions solution was first prepared by mixing the appropriate amount of Pb(II) nitrate salt in distilled water. The removal experiments were conducted by mixing 0.25 g of [MTMSPI][TS] with 12 mL of 200 mg L^−1^ aqueous Pb(II) ions solution in propylene bottles. The propylene bottles were shaken in the orbital shaker with a constant agitation speed (400 rpm) for 30 min at room temperature. The mixture was then separated through centrifugation at 4000 rpm for 15 min to obtain a clear aqueous solution. This clear aqueous solution was analyzed for concentration of Pb(II) ions using AAS. The same experimental procedures were applied to study the performance of activated silica and Si-TS-SSIL to remove Pb(II) ions from the aqueous solution.

pH studies were carried out at various pH values ranging from 2 to 9. 0.25 g of the Si-TS-SSIL extractant, which was mixed with 12 mL of 200 mg L^−1^ aqueous Pb(II) ions solution in propylene bottles. The mixture was shaken at 400 rpm at room temperature. The final concentration of Pb(II) ions was analyzed using AAS and the removal efficiency was then calculated.

Based on the results of the effect of the [MTMSPI][TS] dosage in Si-TS-SSIL pH on the removal efficiency, the experiments for the removal of Pb(II) ions from the aqueous solution were proceeded further using Si-TS-SSIL (mass ratio of [MTMSPI][TS](in g):activated silica gel (in g) = 0.2:1) at pH 6 with the same initial concentration of aqueous Pb(II) ions solution (200 mg L^−1^), but with different contact times ranging from 2 min to 250 min to determine the equilibrium time for the purpose of adsorption isotherm study. The results of the removal efficiency showed that the equilibrium time, *t_eq_*, was 120 min. Thus, to obtain the data for adsorption isotherm study, the experiments for the removal of Pb(II) ions from the aqueous solution were carried out using different initial concentrations of aqueous Pb(II) ions solution, but with the contact time, i.e., 120 min, at room temperature. In this adsorption isotherm study, different initial concentrations of aqueous Pb(II) ions solution were 10, 30, 50, 100, 150, and 200 mg L^−1^.

In the experiments for the removal of Pb(II) ions from aqueous solution, the removal efficiency and removal capacity of extractant were calculated using Equations (6) and (7), respectively [41].
(6)$$\mathrm{Removal}\;\mathrm{efficiency}\;(\%)=\frac{C_{o}-C_{e}}{C_{o}}\times 100$$
(7)$$\mathrm{Removal}\;\mathrm{capacity}\;(q_{e})=\frac{\left(C_{o}\;-\;C_{e}\right)}{m}\times V$$ where *q_e_* (removal capacity) represents the amount of Pb(II) ions removed at equilibrium (mg g^−1^), and *C_o_* and *C_e_* (mg L^−1^) are the initial and equilibrium concentrations of Pb(II) ions in the aqueous solution, whereas *V* (L) and *m* (g) represent the volume of the aqueous lead solution and mass of the extractant, respectively.

### 3.4. Adsorption Isotherm and Kinetics Models

The adsorption isotherms for Pb(II) ions were obtained via removal for 120 min, which was sufficient to reach equilibrium. The effect of the initial concentration of Pb(II) ions (10–200 mg L^−1^) was investigated. The adsorption isotherm and kinetics models were used to determine the ideal fitting in this removal study. Adsorption isotherm models such as Langmuir and Freundlich are used to evaluate and compare the removal capacities of the extractants for the removal of Pb(II) ions in an aqueous solution, whereas for the kinetics studies, the pseudo first and second order models were applied. All models were used to process numerous experimental equilibrium data and to verify which model represents the best fit for the obtained data. The parameters obtained from the Langmuir and Freundlich models provide crucial information about the surface properties, the affinity of the adsorbent, and the extraction mechanism [42].

## 4. Conclusions

In this work, the newly ionic liquid, [MTMSPI][TS], and the functionalized solid supported ionic liquid, Si-TS-SSIL, were successfully synthesized. The formation of an [MTMSPI][TS] ionic liquid has been confirmed with the results from the characterization analyses such as NMR, FTIR, IC, moisture content, and TGA. Meanwhile, the formation of a chemical bond between [MTMSPI][TS] and activated silica gel in the newly synthesized Si-TS-SSIL, i.e., Si-O bond, was confirmed by solid-state NMR analysis. The results of solid-state NMR analysis have shown that an attempt to chemically immobilize [MTMSPI][TS] on activated silica gel to produce Si-TS-SSIL was successful.

The optimum removal efficiency of Pb(II) ions from the aqueous solution was achieved at a contact time of 120 min with the highest removal efficiency of 87% using Si-TS-SSIL that has an [MTMSPI][TS]:activated silica gel mass ratio of 0.2:1. The equilibrium data was fitted with the Langmuir and Freundlich isotherm models whereby the best correlation was obtained by the Freundlich model. The uptake of Pb(II) ions using the Si-TS-SSIL extractant was well described by the kinetics study, which pointed towards the pseudo-second order kinetic model, suggesting the removal process was controlled by the chemisorption process.

The novelty of this study is the synthesis of the [MTMSPI][TS] ionic liquid and the chemical immobilization of the [MTMSPI][TS] ionic liquid onto the surface of a solid support, i.e., activated silica gel, producing Si-TS-SSIL extractant for the removal of Pb(II) ions from an aqueous solution. This work is crucial to ensure that the newly synthesized extractant (Si-TS-SSIL) is effective in removing Pb(II) ions metal ions from aqueous solutions, which would be applicable for other types of metal ions, as well as provide the precise characterization of the newly synthesized [MTMSPI][TS] ionic liquid.

## Figures and Tables

**Figure 1 molecules-28-00830-f001:**
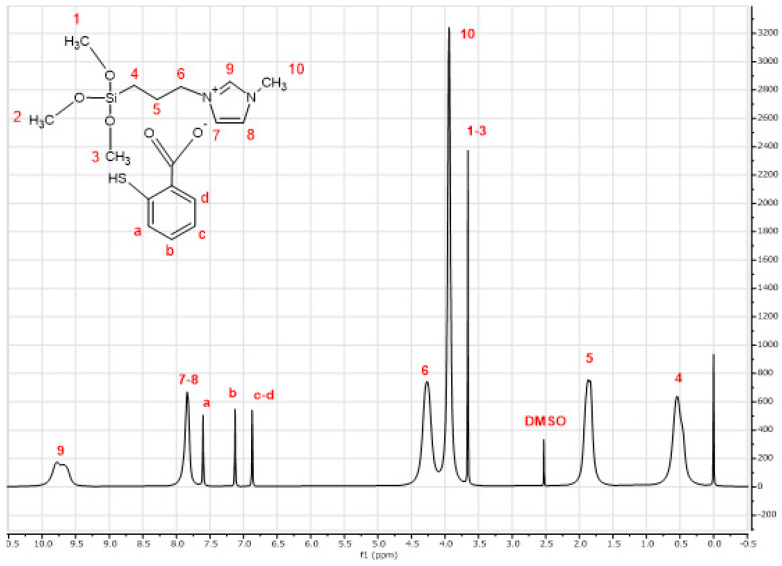
^1^H-NMR spectra of the newly synthesized 1-methyl-3-(3-trimethoxysilylpropyl) imidazolium thiosalicylate ([MTMSPI][TS]) ionic liquid.

**Figure 2 molecules-28-00830-f002:**
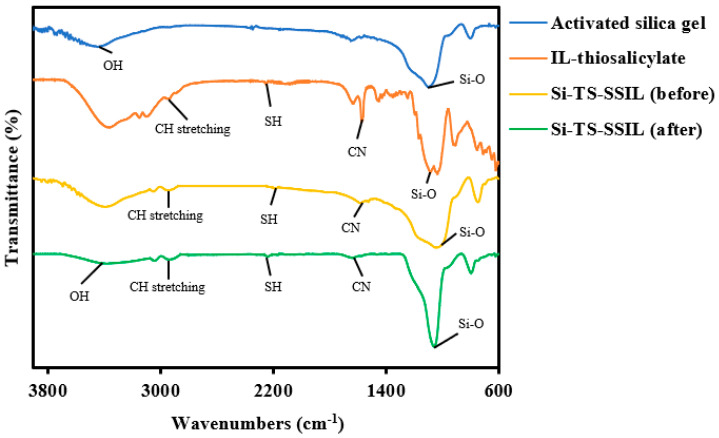
FTIR spectra of the newly synthesized 1-methyl-3-(3-trimethoxysilylpropyl) imidazolium thiosalicylate ([MTMSPI][TS]) ionic liquid, activated silica gel, and Si-TS-SSIL before and after the adsorption process.

**Figure 3 molecules-28-00830-f003:**
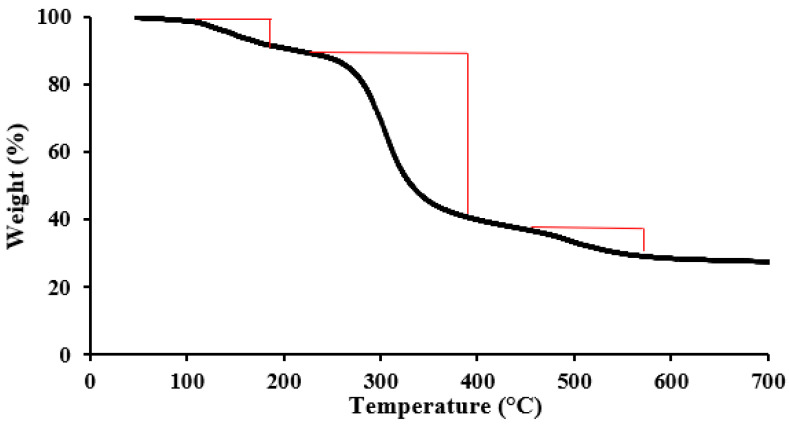
TGA curve of [MTMSPI][TS] ionic liquid.

**Figure 4 molecules-28-00830-f004:**
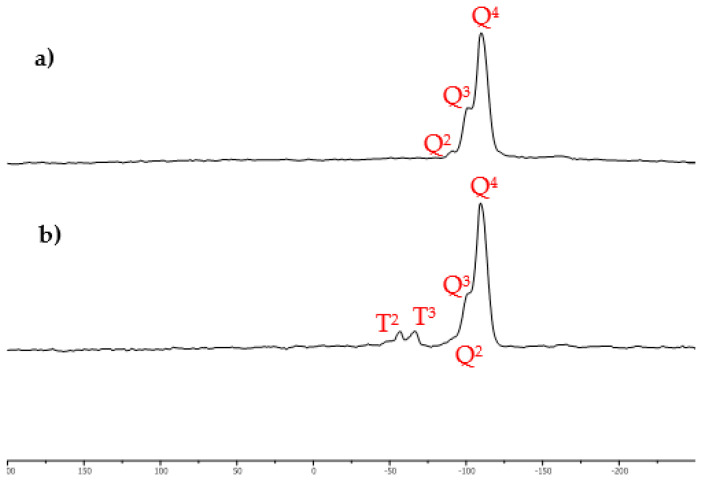
^29^Si NMR spectra of (**a**) pure silica and (**b**) Si-TS-SSIL (modified activated silica gel with [MTMSPI][TS] ionic liquid).

**Figure 5 molecules-28-00830-f005:**
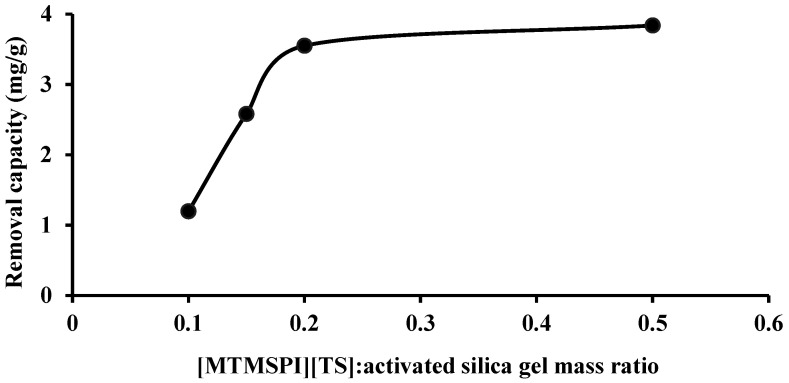
The effect of different [MTMSPI][TS]:activated silica gel mass ratios on the removal process of Pb(II) ions onto Si-TS-SSIL.

**Figure 6 molecules-28-00830-f006:**
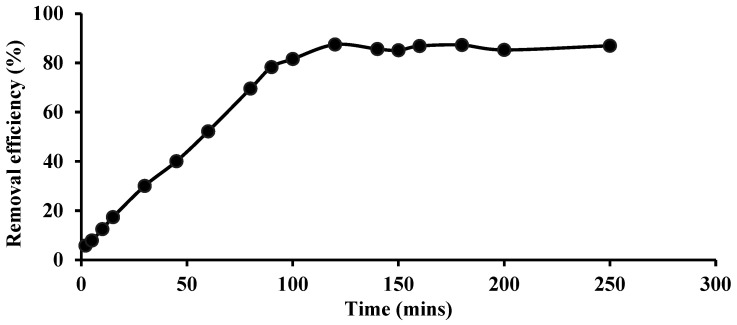
Effect of contact time on Pb(II) ions removal efficiency using Si-TS-SSIL.

**Figure 7 molecules-28-00830-f007:**
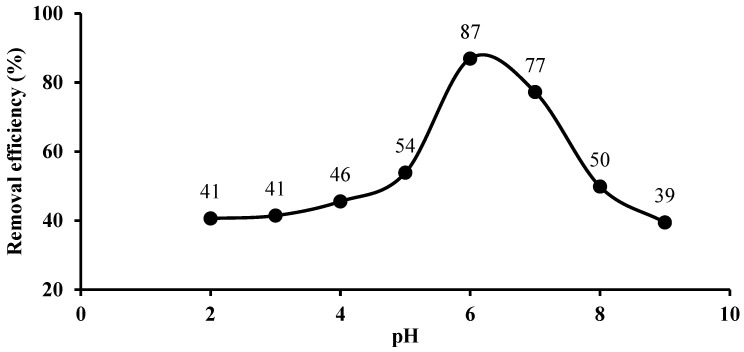
Effect of pH on the removal of Pb(II) ions from aqueous solution using Si-TS-SSIL.

**Figure 8 molecules-28-00830-f008:**
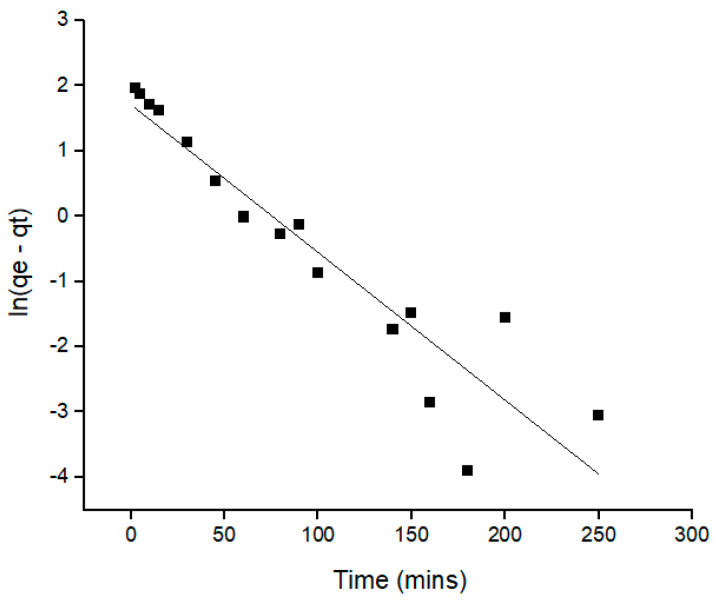
Pseudo-first order kinetic plots for the removal of Pb(II) ions onto Si-TS-SSIL.

**Figure 9 molecules-28-00830-f009:**
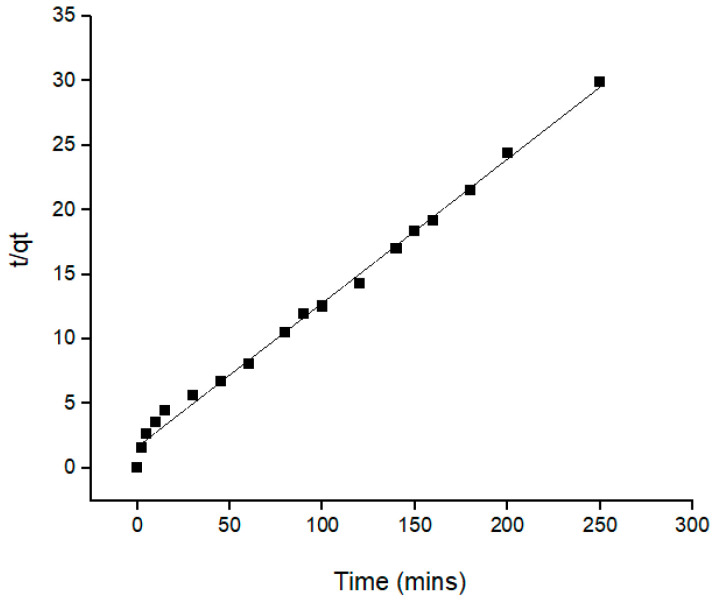
Pseudo-second order kinetic plots for the removal of Pb(II) ions onto Si-TS-SSIL.

**Figure 11 molecules-28-00830-f011:**
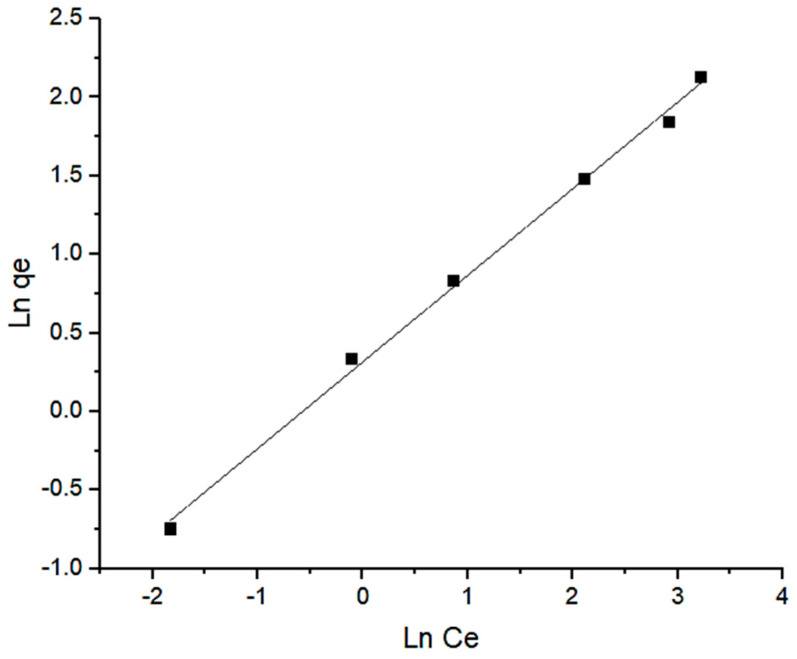
Freundlich isotherm for the removal of Pb(II) ions from aqueous solution by the Si-TS-SSIL.

**Figure 12 molecules-28-00830-f012:**
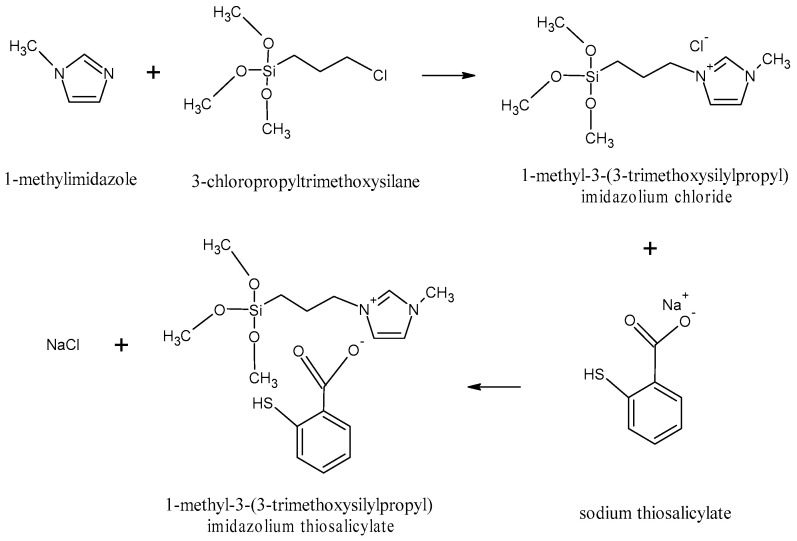
The synthesis route for the formation of [MTMSPI][TS] ionic liquid.

**Figure 13 molecules-28-00830-f013:**
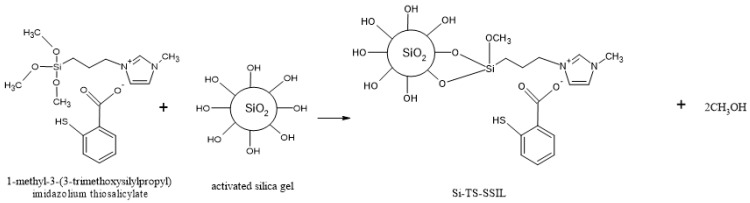
Synthesis route to produce Si-TS-SSIL extractant.

**Table 1 molecules-28-00830-t001:** Chloride content and the purity of the newly synthesized [MTMSPI][TS] ionic liquid.

Chloride Content (ppm)	Concentration (%)	Purity (%)
9.08	0.0908	99.91

**Table 2 molecules-28-00830-t002:** The results of BET analyses for activated silica gel and Si-TS-SSIL (before and after washing with dichloromethane).

Sample	BET Surface Area (m^2^/g)	Pore Volume (cm^3^/g)	Pore Diameter (nm)
Activated silica gel	275.8132	0.7021	11.5176
Si-TS-SSIL (before washing with dichloromethane)	225.6244	0.5758	9.9599
Si-TS-SSIL (after washing with dichloromethane)	232.6600	0.5793	10.2075

**Table 3 molecules-28-00830-t003:** Removal capacity of activated silica gel, [MTMSPI][TS], and Si-TS-SSIL control samples.

Extractant	Removal Capacity (mg/g)
Activated silica gel	1.370
[MTMSPI][TS]	5.801
Si-TS-SSIL	5.285

**Table 5 molecules-28-00830-t005:** Parameters of Langmuir and Freundlich isotherms for the removal of Pb(II) ions by Si-TS-SSIL extractant.

Extractant	Langmuir Isotherm			Freundlich Isotherm		
Si-TS-SSIL	*K_L_* (L/mg)	*q_m_* (mg/g)	*R^2^*	*K_F_* (mg/g)	*n*	*R^2^*
	0.1626	9.3729	0.9102	0.8592	1.8149	0.9961

**Table 6 molecules-28-00830-t006:** Comparison of removal efficiency for Pb(II) removal using Si-TS-SSIL extractant from aqueous solution and wastewater containing 10 mg L^−1^ of Pb(II) ions.

Extractant	Source	Removal Efficiency (%)
Si-TS-SSIL	Aqueous solution	99.33
	Wastewater	80.6

## Data Availability

All relevant data are contained in the present manuscript.

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
