# Peer review of "Synthesis and Characterization of Novel Thiosalicylate-based Solid-Supported Ionic Liquid for Removal of Pb(II) Ions from Aqueous Solution"

_molecules, 2023, doi:10.3390/molecules28020830_

Round 1

Reviewer 1 Report

The title of the manuscript is interesting and the authors used a novel adsorbent for their work. Though the manuscript is interesting, it needs to add some positive aspects. Without improving by the following comments, the manuscript may be rejected.

1- The abstract should be shortened. For instance, the abbreviation name of analysis is enough and the authors should remove the complete name of analyses.

2- The introduction needs to be modified by recent articles. Also, several aspects of this work should be included in the introduction, for example, the difference between various methods of treatment in heavy metal removal. Also, the reason for removing pollutants from wastewater, etc. The following papers can be cited and used:

https://doi.org/10.1016/j.apenergy.2021.117603; https://doi.org/10.1021/acs.est.8b02213; https://doi.org/10.1016/j.scitotenv.2019.05.060; https://doi.org/10.1016/j.micromeso.2022.112248; https://doi.org/10.3390/met12071160

https://doi.org/10.1016/j.watres.2021.117491; https://doi.org/10.1016/j.jelechem.2022.117062

https://doi.org/10.1016/j.jclepro.2022.135262

3- What is the innovation of this work? The authors should mention this point at the end of the introduction.

4-All chemicals and devices should be characterized (e.g., model, company, country) in the materials and method section.

5-FTIR analysis after the process should be added to the manuscript.

6- Compounds should be highlighted on the NMR figure.

7- More analyses are required for determining the characterization of the materials such as SEM, or XRD

7- Why the sorption capacity is very low? Please justify.

8- For Eqs. 6 and 7, relevant references should be added to the manuscript.

9- What is the impact of ionic strength? Did authors calculate this important factor?! Also, the impact of pH has not been investigated!

10- What about the real sample of wastewater?

11- Several references are old and should be replaced.

Author Response

Thank you.

Reviewer 2 Report

In the present manuscript, authors have reported the removal of Pb(II) ions  from water by adsorption on solid supported ionic liquids. The subject is interesting however there are certain quarries which must be satisfied before final publication:

 1) There are many typos/grammatical mistakes in the manuscript e.g Page 1 Line 11: “The main objectives of this study are to synthesis… ” needs to be revised as “The main objective of this study is to synthesize…” Authors should carefully revise the manuscript.

2) Page 1 Line 30 “with variation initial concentration of Pb(II) ions…”  needs revision

3) Page 1 Line 31-32 needs revision

4) First two sentences of introduction should be revised.

5) Page 2 Line 60: “solvents which is (are) carcinogenic….”

6) Page 2 Line 67: “Ionic liquids (Ionic liquid), mainly consists of an organic…”

7) Page 2 Line 73: “as replacing ILs with common volatile organic solvents…” I think vice versa is correct.

8) Page 2 Line 80: “costly” may be replaced with ”high cost”

9) Page 2 Line 84:is being studied extensively by many researchers.” Add relevant references.

10) Mention of an important method for removal of metal ions from waste water using plant biomass is missing in the introduction section.  Here is a relevant reference for more detail:

Naseem, K.; Farooqi, Z. H.; Begum, R.; Rehman, M. Z.; Shahbaz, A.; Farooq, U.; Ali, M.; Rahman, H. M. A.; Irfan, A.; Al-Sehemi, A. G. Removal of Cadmium (II) from Aqueous Medium Using Vigna radiata Leave Biomass: Equilibrium Isotherms, Kinetics and Thermodynamics. Zeitschrift fur Physikalische Chemie 2019, 233, 669-690.

11)  Page 3 Lines 103-104: “The kinetics studies for the removal of Pb(II) ions was carried out to determine the adsorption isotherms.” Adsorption isotherms cannot be determined using kinetic studies. Needs revision

12) Table 2 indicates that activated silica gel has larger surface area, pore volume and pore diameter than Si-TS-SSIL but adsorption of Pb(II) ions is comparatively poor. How authors justify it?

13) Page 9 Line 247 needs revision.

Author Response

Thank you. 

Round 2

Reviewer 1 Report

After checking, I think the manuscript can be published in this journal.

Reviewer 2 Report

Authors have incorporated all the suggested changes in the revised manuscript. The manuscript is recommended for publication.